# The recombination landscape of introgression in yeast

**Enrique J. Schwarzkopf**, **Nathan Brandt**, **Caiti Smukowski Heil***

Department of Biological Sciences, North Carolina State University, Raleigh, North Carolina, United States of America

* cheil@ncsu.edu

## Abstract

Meiotic recombination is an evolutionary force that acts by breaking up genomic linkage, increasing the efficacy of selection. Recombination is initiated with a double-strand break which is resolved via a crossover, which involves the reciprocal exchange of genetic material between homologous chromosomes, or a non-crossover, which results in small tracts of non-reciprocal exchange of genetic material. Crossover and non-crossover rates vary between species, populations, individuals, and across the genome. In recent years, recombination rate has been associated with the distribution of ancestry derived from past interspecific hybridization (introgression) in a variety of species. We explore this interaction of recombination and introgression by sequencing spores and detecting crossovers and non-crossovers from two crosses of the yeast *Saccharomyces uvarum*. One cross is between strains which each contain introgression from their sister species, *S. eubayanus*, while the other cross has no introgression present. We find that the recombination landscape is significantly different between *S. uvarum* crosses, and that some of these differences can be explained by the presence of introgression in one cross. Crossovers are significantly reduced in heterozygous introgression compared to syntenic regions in the cross without introgression. This translates to reduced allele shuffling within introgressed regions, and an overall reduction of shuffling on most chromosomes with introgression compared to the syntenic regions and chromosomes without introgression. Our results suggest that hybridization can significantly influence the recombination landscape, and that the reduction in allele shuffling contributes to the initial purging of introgression in the generations following a hybridization event.

## Author summary

Mating between different species (i.e., hybridization) can introduce novel and distinct genetic variation in a genome. The persistence and distribution of this variation introduced from hybridization is influenced by many processes, including recombination, which occurs during gamete development in all sexually reproducing organisms and shuffles alleles between homologous chromosomes. Here, we set up crosses between 1) strains of yeast without variation from hybridization, and 2) strains of yeast with variation introduced from hybridization, in order to understand the interaction between recombination

**Data availability statement:** Sequences for the parental strains can be found on NCBI SRA (SRR1119189, SRR1119180,

SRR1119199, SRR1119200) (Almeida et al., 2014 doi:10.1038/ncomms5044). Sequencing of the tetrads is deposited at NCBI SRA under Project PRJNA1061120. Scripts are available in the github repository: https://github.com/ejschwarzkopf/CO-NCO.

**Funding:** This work was supported by a National Institute of General Medical Sciences (https://www.nigms.nih.gov/) R35GM142849 to CSH. The funders had no role in study design, data collection and analysis, decision to publish, or preparation of the manuscript.

**Competing interests:** The authors have declared that no competing interests exist.

and variation introduced from hybridization. We observe that DNA sequence differences in regions derived from hybridization suppress the reciprocal exchange of genetic material (known as crossovers), and most recombination events are repaired through an alternate resolution known as non-crossovers. Variation from hybridization changes the amount of allele shuffling locally and across chromosomes, likely helping to facilitate the elimination of genetic variation from hybridization after hybridization occurs.

## Introduction

Recombination is the exchange of genetic material between homologous chromosomes during meiosis and is a staple of eukaryotic sexual reproduction. While the processes involved in recombination are largely conserved [1], recombination rates vary between sexes, populations, and species [2,3]. Recombination rates also vary along the genome, with conflicting patterns of enriched or depleted recombination in promoter regions and punctate or dispersed recombination depending on the species [4–7]. These patterns in recombination can affect pairing of alleles after meiosis–in other words, the shuffling of alleles–in a population. Much of the evolutionary advantage of recombination is understood to originate from its role in shuffling alleles, both through independent assortment of chromosomes during meiosis and through exchange between homologous chromosomes, which increases the number of different allele combinations segregating in a population. The increase in allele combinations can reduce selection interference–the effect that genetically linked sites have on the evolutionary fate of either beneficial or deleterious alleles [8–11].

How much allele decoupling is produced by recombination will depend on the type of recombination event. Each recombination event begins with the severing of both strands of a sister chromatid of one of the homologous chromosomes in what is referred to as a meiotic double-strand break (DSB) [12]. A meiotic DSB is repaired as a crossover (CO), which results in the reciprocal exchange of genetic information between homologous chromosomes, and/or as a non-crossover (NCO) gene conversion—where a small segment (typically 100–2000 bp) of a chromosome is replaced by a copy of its homolog [13–16]. The majority of COs in most organisms are generated via the class I pathway, also known as the ZMM pathway (named for the *ZIP1/2/3/4*, *MER3* and *MSH4/5* genes); COs formed via this pathway are controlled in number (crossover assurance, homeostasis) and distribution (crossover interference) [17–20]. A minority of COs are formed via the class II pathway (*MUS81* dependent) [21–23]. NCOs are formed either via the synthesis dependent strand annealing (SDSA) pathway or through the dissolution of Holliday junctions [24–26]. COs generally produce more allele shuffling, and therefore degrade linkage faster than NCOs. However, NCOs can occur in regions where COs are typically suppressed, like centromeres and inversions [27–33]. NCOs are also crucial to reducing linkage within coding regions and, unlike COs, result in 3:1 allele ratio in the meiotic product at heterozygous sites, potentially changing allele frequencies [34].

Variation in the number and distribution of COs and NCOs, and their respective associated effects on linkage, have important implications for molecular evolution. Recombination has long been appreciated to play a role in the distribution of various genomic features including nucleotide diversity. Nucleotide diversity has a positive correlation with recombination rate in a number of species, interpreted to result from selective sweeps and background selection removing genetic variation in regions of low recombination [35–37]. Similarly, recombination breaking up genetic associations is particularly notable in the context of interspecific hybridization. In first-generation (F$_1$) hybrids, the hybrid genome is heterozygous for each parental species. If the hybrids then back-cross to one of the parental species, recombination will produce

genomes that are a mosaic of genetic information from the two species with a minor contribution from the species that was not backcrossed to (introgression) [38]. When each population has evolved alleles that are deleterious when present in the background of the other population (the Dobzhansky-Muller hybrid incompatibility model) we expect introgressed regions with low rates of recombination to be quickly purged from the population, as the accumulation of incompatible alleles incurs a steep fitness cost. In contrast, when introgressed regions have high recombination rates, the break up of genetic associations will reduce selective interference between the incompatible alleles and their surrounding haplotypes, allowing for neutral and beneficial alleles brought in with the introgression to escape the fate of neighboring incompatibilities [39–44]. This theory is supported empirically through enrichment of introgressed segments in regions of higher recombination in a number of organisms including Mimulus, maize, butterflies, swordtail fish, stickleback, and humans [43,45–49].

This positive correlation between introgressed ancestry and recombination is emerging as a nearly ubiquitous pattern (though see [50–52]), however, it is unclear how these observations relate to the known effect of sequence divergence on DSB repair. Introgression, particularly between highly diverged species, can have low sequence similarity with the genomic region it is replacing. While DSBs are not sensitive to heterozygosity, the repair of DSBs into COs or NCOs is dependent on sequence differences. Mismatch repair proteins function during meiosis to detect and repair mismatches, which ensures COs are occurring between homologous chromosomes and at equivalent positions to prevent ectopic recombination [53–56]. CO events are known to decrease as sequence divergence increases due to heteroduplex rejection of diverged sequences [57–62]. Rejected events may be repaired as NCO through the SDSA pathway or through dissolution of double Holliday junctions [24,61,63]. Given that heterozygous introgression will have divergent sequences, we expect a decrease in COs, and possibly an increase in NCOs as DSBs fail to be repaired as COs in heterozygous introgression.

To help us understand this interaction of introgression and recombination, and identify patterns in CO and NCO in closely related populations, we utilized the budding yeast *Saccharomyces*. Yeasts provide an excellent opportunity to study DSB repair, as we can readily isolate and collect all four meiotic products of a given meiosis and detect both CO and NCO events (Fig 1A) [28,64–67]. Recombination rates vary between strains of *S. cerevisiae* [68,69] and between *S. cerevisiae* and its sister species *S. paradoxus* [67,70]. Strains of different *Saccharomyces* species have often hybridized with other species and carry introgressed DNA from these events [71–77].

In this study, we look at patterns of recombination and introgression by crossing two pairs of Holarctic *Saccharomyces uvarum* strains. One pair of strains was isolated from natural environments in North America and the other pair was isolated from European fermentation environments [72]. The *S. uvarum* strains isolated from European fermentation environments each carry introgression from their sister species, *Saccharomyces eubayanus*, which is approximately 6% divergent from *S. uvarum* [72,78,79]. The diploid $F_1$ genome of these strains is heterozygous for nine different introgressions which make up approximately 10% of the genome (Fig 1B). The strains from the North American cross do not carry *S. eubayanus* introgression, thus allowing us to assess the impact of introgression on the recombination landscape. We obtained whole genome sequencing data from individual meiotic events from the first offspring generation of each cross and used this data to detect CO and NCO events along the genome (Fig 1A). From these maps, we aim to understand (i) how patterns of CO differ between closely related strains, (ii) how regions of introgression differ in their CO and NCO patterns, and (iii) how these different patterns affect shuffling of alleles locally and at the chromosome level. Understanding these objectives will provide us novel insights into how introgression impacts the recombination landscape.

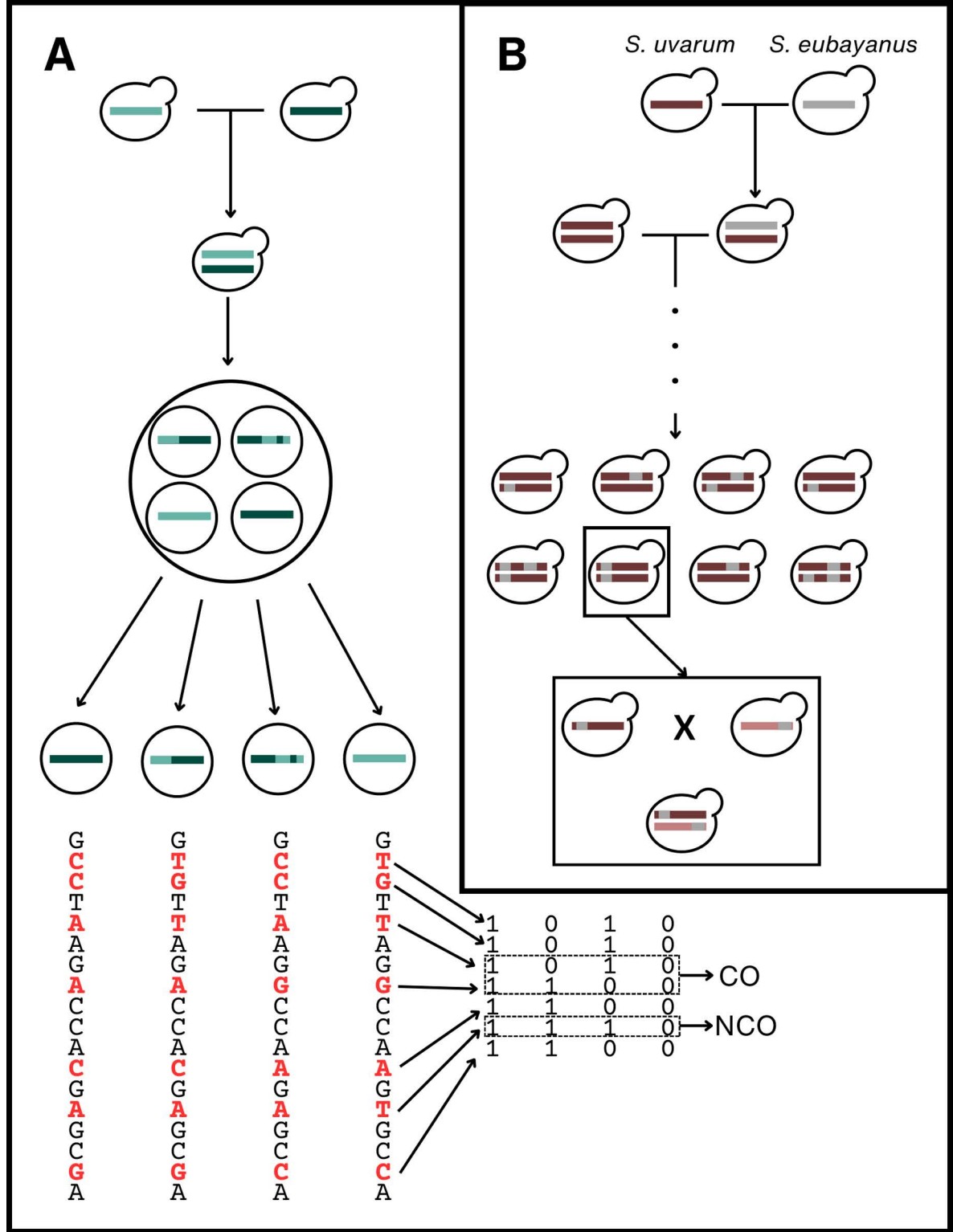

**Fig 1. Overview of *Saccharomyces uvarum* strains and experimental design.** (A) Visual representation of how our crosses were conducted. Haploid yeast from two genetically distinct parental strains are mated to produce a heterozygous diploid. Meiosis is induced and the resulting meiotic products (tetrads) are manually dissected, and each haploid meiotic product is grown mitotically to obtain enough material for DNA extraction and whole genome sequencing. We then call SNPs on the resulting sequences and retain loci with fixed differences between parents. These loci are then coded as 1 or 0 depending on the parent of origin and the CrossOver software detects

COs and NCOs. (B) A schematic of how introgression likely arose in the strains sampled from fermentation environments. These introgressions are likely due to *S. eubayanus* hybridizing with *S. uvarum* at some point in the past, resulting in F1 hybrids that then potentially crossed with other *S. uvarum* individuals for some number of generations. Eventually, the *S. eubayanus* ancestry was degraded in the population of *S. uvarum* until the introgressions we observe today remained, potentially segregating in the population. A similar process likely happened in each of the parental strains we utilized, but with different introgressions remaining in each strain. We crossed haploid individuals from two parental strains, resulting in a diploid that is heterozygous for each introgression.

## Results

### The recombination landscape differs between closely related crosses

We isolated and sequenced products of 48 meioses (192 haploid spores) for each of two crosses of *S. uvarum*, a cross between strains isolated from North America (natural cross) and a cross between strains isolated from Europe (fermentation cross) (a total of 384 spores). We detected COs across the 16 nuclear chromosomes of *S. uvarum*. Genomewide, we found 82.54 COs/meiosis (SE 1.5; 0.72 cM/kb) on average in the natural cross and 63.66 COs/meiosis (SE 1.9; 0.55 cM/kb) in the fermentation cross (S1 and S2 Tables). The number of COs per meiosis in the natural and fermentation crosses are slightly higher than those of particular strains of *S. paradoxus* (54.8) and *S. cerevisiae* (76.5) respectively [67], though there is known variability in CO counts per meiosis for *S. cerevisiae* (CO: 90.5, 76.3, 73; [28,62,67]).

To further explore the differences in recombination landscapes between our crosses, we split the genome into 20kb, non-overlapping windows, and obtained CO, NCO, and marker counts for each region (Figs 2 and S1). Detection of both CO and NCO events depends on sequence differences between parental strains; this is especially important for NCO events due to their short conversion tracks. Because we utilized closely related wild isolates, marker density is variable across the genome. To evaluate the effect of marker density on our ability to detect CO and NCO events, we downsampled markers in the introgressions of the fermentation cross, where marker density is high, and found a significant effect of marker density on NCO count and tract length for most of the regions (S2–S10 Figs and S3 Table). Conversely, CO rates were generally unaffected by marker density–with the exceptions of portions of the introgressed regions of chromosomes 7, 9, and 14 (S3 Table), which have regions of high and low sequence similarity (Fig 3). Because of how correlated marker density, NCO count, and introgression are, disentangling their relationships is complicated. For this reason, we applied an especially stringent correction for marker density–we used a previously published simulation based method for correction [33,67] and took the average NCO tract length (550 bp) from introgressions, where our marker density is highest, as the correction factor. For each window across the genome, we established our expected probability of detecting an NCO of that length and divided our observed NCO count by our probability of detecting an NCO event (see Methods). After the correction we found modest–but significant–genomewide correlation between our crosses for both COs (Spearman's correlation: 0.27; p<0.0001; S4 and S5 Tables) and NCOs (Spearman's correlation: 0.13; p=0.0018).

We hypothesized that differences in the recombination landscape between crosses might be impacted by the presence of heterozygous introgression from *S. eubayanus* in the fermentation cross. To explore this possibility, we separated the 20kb windows into introgressed and non-introgressed windows (based on whether they overlapped with an introgressed region). We will refer to introgression in the fermentation cross as "introgression" and use the term "introgressed region" to refer generally to the syntenic region, regardless of which cross we are focusing on. We find that introgressions tend to have lower CO counts and higher NCO counts in the fermentation cross when compared to syntenic

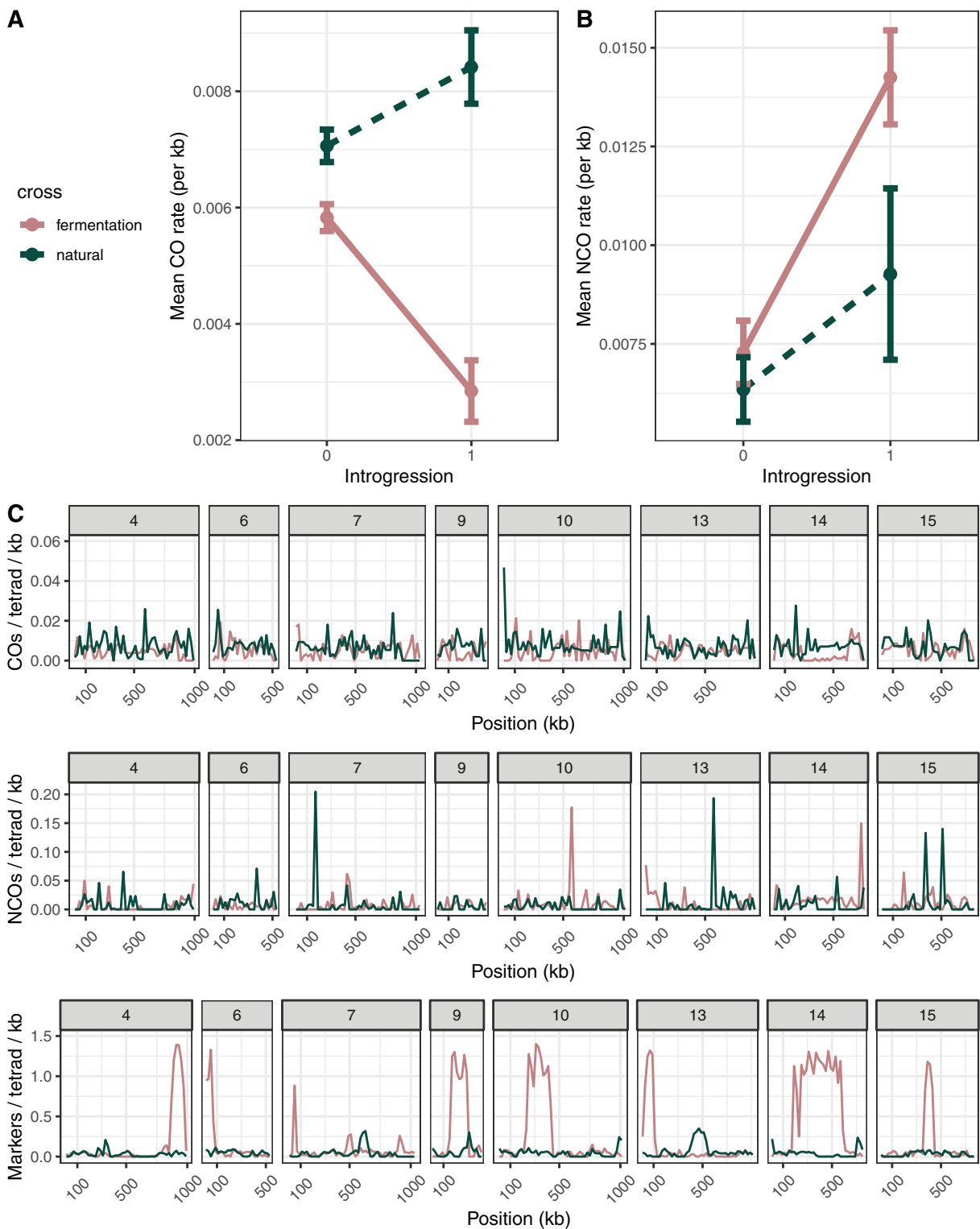

**Fig 2. The recombination landscape is significantly different in introgression.** (A) Mean CO/kb and (B) NCO/kb by cross and introgression (0 denotes intervals without introgression; 1 denotes introgression present in the fermentation cross. While the natural cross does not contain introgression, the region where introgression is present in the fermentation cross was compared to its syntenic region in the natural cross). NCO counts are corrected for marker resolution. Error bars represent the standard error around the mean. (C) *S. uvarum* chromosomes containing introgressions split into 20kb, non-overlapping windows. CO, NCO, and SNP counts are reported for both crosses (fermentation

and natural). Shaded regions denote introgressed regions. CO counts are smoothed when the true location of the CO split could be in one of multiple windows. NCO counts are corrected for marker resolution.

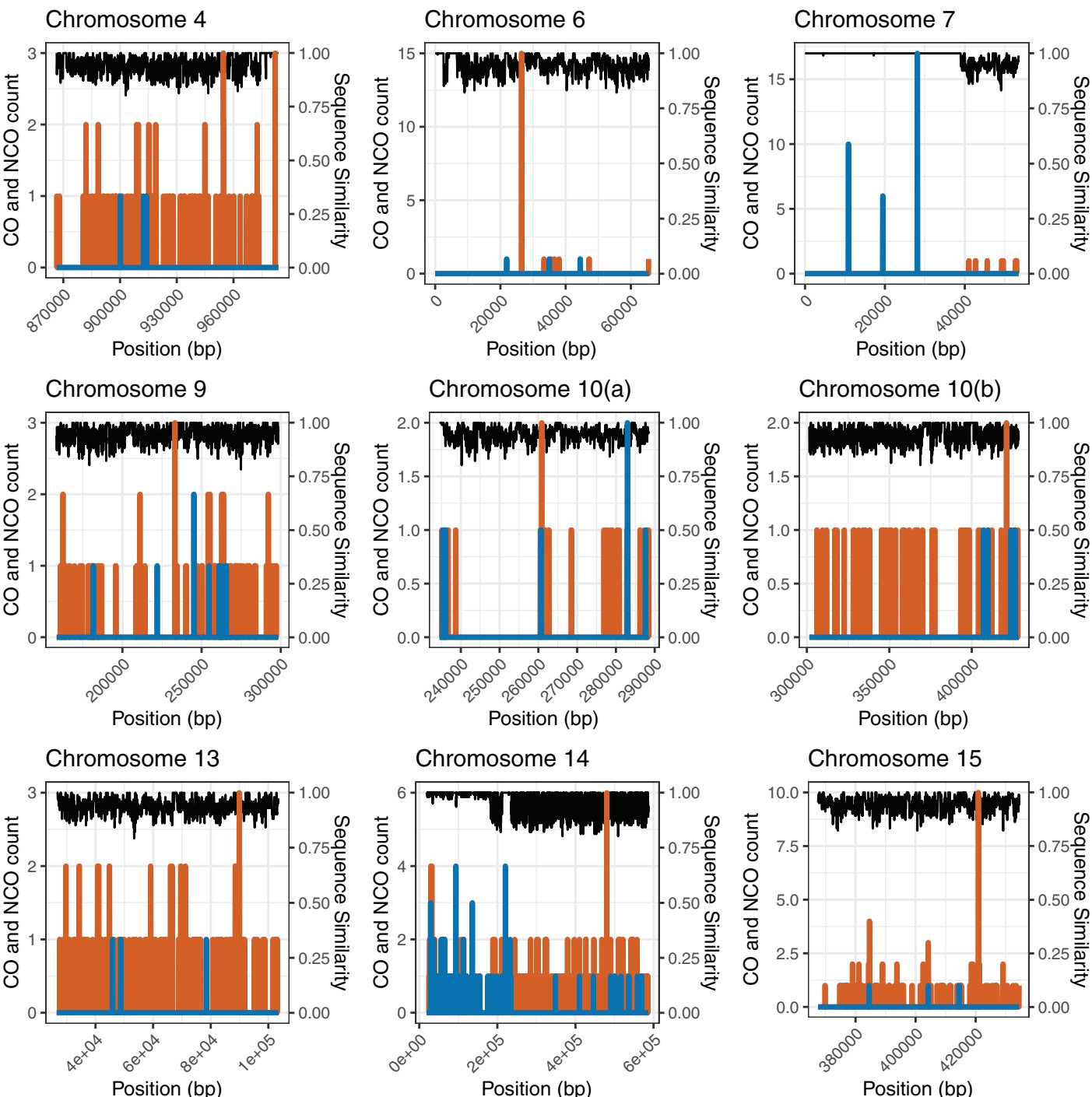

**Fig 3. CO, NCO, and sequence similarity in 101 bp sliding windows with 50 bp overlaps of fermentation cross introgression.** CO counts are shown in blue, the depth of NCO tracts are shown in orange, and the proportion of expected homologous bases between the two fermentation strains is shown in black.

regions in the natural cross (Fig 2A and 2B; Wilcoxon rank sum test; CO p-value<$10^{-5}$, NCO p-value<$10^{-5}$). The fermentation cross has fewer COs than the natural cross overall, but the difference is greater in the introgressed regions. We also found that introgressed regions in both the natural and fermentation cross show significantly more NCOs than non-introgressed regions of the same cross, whereas COs were more frequent in introgressed regions of the natural cross but less frequent in introgressed regions of the fermentation cross (Wilcoxon rank sum test; natural: CO p-value=0.007, NCO p-value=0.006; fermentation: CO p-value<$10^{-5}$, NCO p-value<$10^{-5}$). Despite the decrease in COs in the introgressed regions of the fermentation cross, both crosses saw an increase in total events (CO + NCO) in introgressed regions (Wilcoxon rank sum test; natural: p-value=0.0005; fermentation: p-value<$10^{-5}$), indicating that these regions may be predisposed to having more recombination events.

To further explore and test possible explanations for the patterns of COs and NCOs in introgressed regions, we constructed linear models for CO and NCO counts. Our model of CO counts showed a significant positive effect of cross (windows from the natural cross showed more COs overall), the interactions between natural cross and introgression (natural cross windows show increased CO rate if they are also in an introgression), and GC content. The CO model also shows a significant negative effect of introgression on the number of COs, reflecting the large decrease in COs in these regions in the fermentation cross (Table 1). Our model of NCO counts showed a similar positive effect of GC content on NCO counts, but showed an opposite significant coefficient for introgression and no significant effect of the interaction between introgression and cross–which was removed from the final model (Table 2). This indicates that GC content still plays an important role in localizing NCOs, and supports our findings that patterns of NCOs in introgressed regions are different to those of COs. The lack of a significant effect of cross indicates that differences in overall NCO rates are not well explained by whether a window is from the natural or fermentation cross. The lack of a significant effect of the interaction between introgression and cross indicates that these differences are better explained by comparing crosses at introgressed and non-introgressed regions separately. This is consistent with our findings in introgressed regions in the natural cross–where there is an increase in NCOs when compared to non-introgressed regions, but that difference is smaller than that of the fermentation cross.

**Table 1. Coefficients of gaussian generalized linear model modeling CO counts per 20kb window.**

|  | Estimate | Std. Error | t value | Pr(>|t|) |
| --- | --- | --- | --- | --- |
| (Intercept) | −9.5157 | 1.4725 | −6.462 | 1.51e-10 |
| Cross (natural) | 1.1603 | 0.3210 | 3.615 | 0.0003 |
| Introgression | −3.0542 | 0.6463 | −4.726 | 2.57e-06 |
| GC | 37.9733 | 3.6846 | 10.306 | <2e-16 |
| Introgression:crossnatural | 4.0776 | 0.9133 | 4.465 | 8.80e-06 |

**Table 2. Coefficients of gaussian generalized linear model modeling NCO counts per 20kb window.**

|  | Estimate | Std. Error | t value | Pr(>|t|) |
| --- | --- | --- | --- | --- |
| (Intercept) | −11.4763 | 4.7810 | −2.4004 | 0.0165 |
| Introgression | 4.3490 | 1.4937 | 2.9115 | 0.0037 |
| GC | 45.2894 | 12.0353 | 3.7631 | 0.0002 |

## Reduced diploid sequence similarity helps explain non-crossover repair of DSBs in introgression

One possible explanation for the decrease of COs and increase of NCOs in introgression is that the reduced sequence similarity is biasing DSBs in the region to be repaired as NCOs rather than COs. We would therefore expect to see NCOs to be negatively correlated with sequence similarity. To evaluate the relationship of sequence similarity to the CO and NCO landscapes in introgressions, we measured diploid sequence similarity (the proportion of bases that are expected to match when we sample one base from each of the two parental strains), NCO depth, and CO count in 101 bp sliding windows with 50 bp overlaps along each of the introgressions. This analysis was done exclusively in the fermentation cross and utilized uncorrected NCO counts. Mismatch repair proteins in *Saccharomyces* seem to suppress COs with very little mismatch in small regions (~350 bp), which informed our window size [60]. We counted the number of NCO tracts that intersect with each window as a measurement of NCO depth, and simply counted the CO events in a given window (Fig 3). We then ran Spearman's correlations and a loess regression along each introgression and found a weak, but often significant (p<<0.001) correlation between NCOs and sequence similarity in the introgression (Table 3), suggesting that repair of double strand breaks is biased towards NCOs when sequence similarity is low. From the loess regression, we can observe an increase in NCO as sequence similarity reduces until about 0.9–0.8 sequence similarity, at which point NCOs level out or reduce (S11 Fig). However, this effect is very weak with respect to the NCO counts, and at low levels of sequence similarity the uncertainty of the regression line is very large. This is primarily driven by the number of windows with no NCOs.

The low CO count in introgression leaves us unable to investigate effects of sequence similarity on CO counts except on chromosomes 7 and 14, where we found significantly higher sequence similarity around COs than around NCOs (Table 4). The introgressions on these two chromosomes are unique in that they contain a highly homologous portion of sequence where the same introgressed sequence is present in both parents, and therefore contain enough COs for us to have power to detect differences between CO and NCO neighborhoods (Fig 3). This also highlights that the general pattern of reduced CO and increased NCO in introgression (Fig 2) is not a result of introgression itself, but of heterozygosity influencing the CO/NCO decision.

## Introgression decreases allele shuffling locally and at the chromosome level

Because NCOs still play a small role in shuffling alleles along the chromosome, we were interested in whether the increase in NCOs of the fermentation cross would supplement the

**Table 3. Spearman's correlations of NCOs to sequence similarity in introgression in the fermentation cross. A cutoff p-value of 0.001 was selected by calculating the Bonferroni corrected α=0.05 for nine comparisons (0.0011) and rounding down.**

| Chromosome | Start | End | Corr | p<0.001? |
|---|---|---|---|---|
| 4 | 866500 | 983774 | −0.1686 | TRUE |
| 6 | 1 | 65500 | −0.0394 | FALSE |
| 7 | 1 | 53500 | −0.2435 | TRUE |
| 9 | 158500 | 298500 | −0.1460 | TRUE |
| 10 | 234500 | 288500 | 0.0073 | FALSE |
| 10 | 301500 | 428500 | −0.1097 | TRUE |
| 13 | 26500 | 103500 | −0.1982 | TRUE |
| 14 | 18500 | 586500 | −0.2300 | TRUE |
| 15 | 367500 | 434500 | −0.2328 | TRUE |

**Table 4. Welch two sample t-test results for differences in sequence similarity between CO-adjacent regions and NCO-adjacent regions per introgression.**

| Introgression | CO mean sequence similarity | CO SE | NCO mean sequence similarity | NCO SE | p-value |
|---|---|---|---|---|---|
| Chromosome 4 | 0.9467 | 0.0148 | 0.9248 | 0.0029 | 0.2835 |
| Chromosome 6 | 0.9663 | 0.0203 | 0.9784 | 0.0020 | 0.5594 |
| Chromosome 7 | 1.0000 | 0.0000 | 0.9838 | 0.0018 | <0.0001 |
| Chromosome 9 | 0.9650 | 0.0102 | 0.9312 | 0.0037 | 0.0084 |
| Chromosome 10 (1) | 0.9671 | 0.0092 | 0.9476 | 0.0066 | 0.0419 |
| Chromosome 10 (2) | 0.9400 | 0.0074 | 0.9276 | 0.0040 | 0.1575 |
| Chromosome 13 | 0.9700 | 0.0161 | 0.9304 | 0.0025 | 0.1292 |
| Chromosome 14 | 0.9893 | 0.0026 | 0.9415 | 0.0018 | <0.0001 |
| Chromosome 15 | 0.9317 | 0.0093 | 0.9252 | 0.0031 | 0.5088 |

lost shuffling from the suppression of COs in the introgressions. To test this hypothesis, we used the measure $\bar{r}$ which accounts for the number and positioning of recombination events to estimate the probability that a randomly chosen pair of loci shuffles their alleles in a gamete [80]. We calculated the average $\bar{r}$ per chromosome and for each introgressed region for each of the two crosses. We observed high levels of shuffling at the chromosome level when compared to humans. The intra-chromosomal component of $\bar{r}$ in humans is 0.0135 in females and 0.0177 in males [80], while our measurements for chromosomes varied between 0.216 and 0.426. We find that most chromosomes do not have a significantly different amount of allele shuffling between the two crosses, even though the natural cross generally has more COs (S6 Table; Bonferroni-adjusted $\alpha = 0.00313$). However, of the six chromosomes with significantly different $\bar{r}$ values, all of them showed more shuffling in the natural cross, and five of the six (chromosomes 4, 9, 10, 14, and 15) contained introgressed regions (Fig 4). Lower $\bar{r}$ is not observed when introgressions are small and near telomeres, while even a small introgression near the center of the chromosome can lead to a large reduction in $\bar{r}$ (as is the case for chromosome 15). Chromosome 12 was the only chromosome without an introgressed region to have significantly different shuffling between crosses, and it also showed more shuffling in the natural cross. While it is unclear what potential mechanism is mediating the difference in shuffling on chromosome 12, we note that the rDNA locus on chromosome 12 is known to differ dramatically in repeat content across strains of *S. cerevisiae* (22–227 copies) [81], and we speculate that differences in rDNA copy number between strains in our crosses could impact shuffling. All of the introgressed regions showed significantly more shuffling in the natural cross, indicating that the large increase of NCOs in the introgressions does not make up for the loss of shuffling from the depletion of COs (S7 Table; Bonferroni-adjusted $\alpha = 0.00556$). This finding indicates that an introgression that is segregating in a population will incur a shuffling cost in heterozygous individuals on top of any other evolutionary effects the introgression may have.

## Discussion

Our study is motivated by understanding recombination rate variation within a species and uncovering potential genetic factors underlying this variation. To investigate this question, we crossed two pairs of *S. uvarum* strains, one pair isolated from natural environments and one pair from fermentation environments, and explored the distribution of recombination events from both crosses. We detected more COs in our natural cross when compared to our fermentation cross, within a similar range of COs per meiosis as previous studies in *S. cerevisiae* and

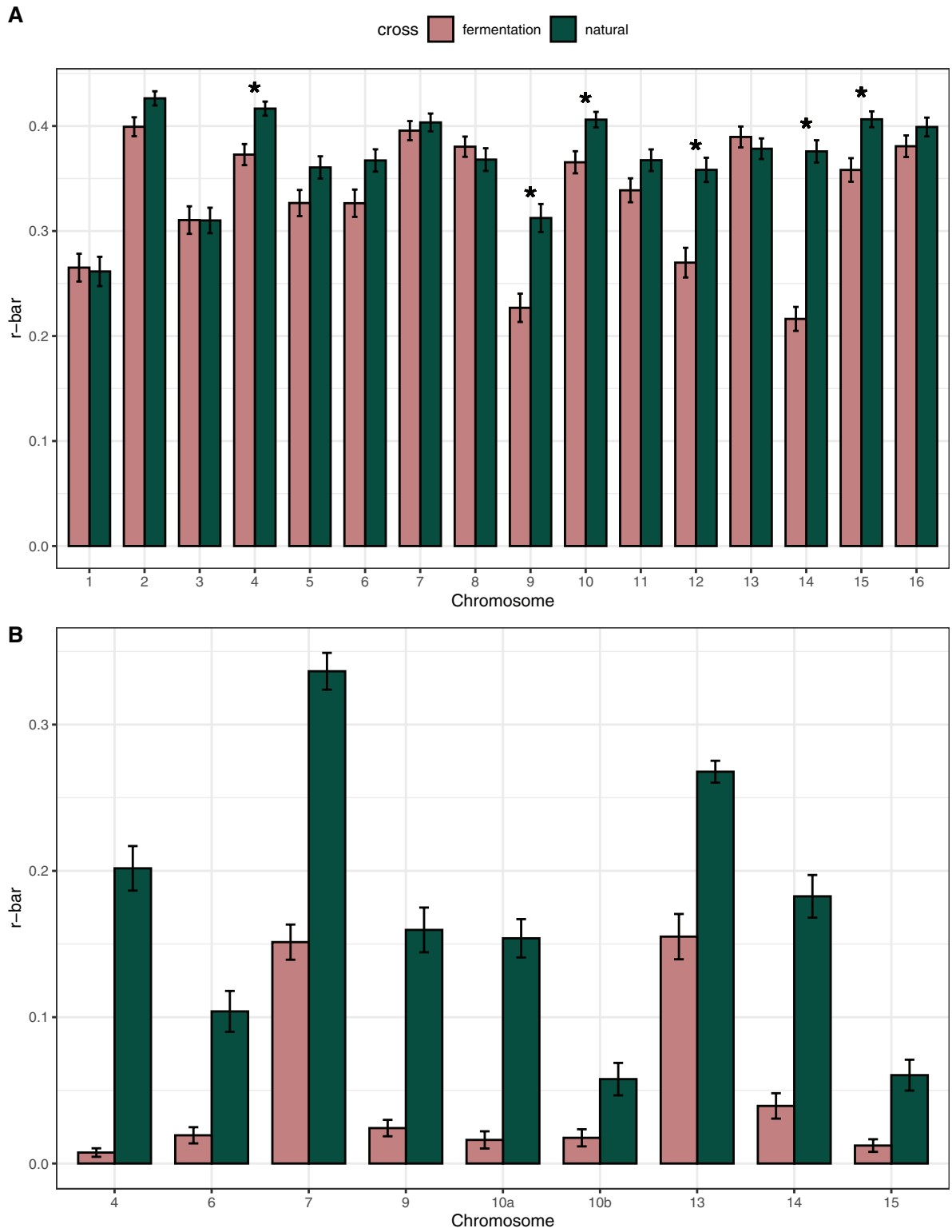

**Fig 4. Introgression decreases allelic shuffling.** Average $\overline{r}$ for each chromosome (A) and for each introgressed region (B). Asterisks indicate a significant difference in chromosome $\overline{r}$ between crosses. All introgressed regions had a significant difference in $\overline{r}$. Error bars indicate standard error around the mean.

*S. paradoxus* [28,67]. Our findings demonstrate significant differences in recombination rate between closely related strains, adding to a body of literature across plants, animals, and fungi that recombination rate can evolve rapidly [11,69,82–88].

We hypothesized that these differences in the recombination landscape between our crosses were in part influenced by introgression, given that heterozygous introgression creates sequence divergence, and that COs in regions of heterozygosity are known to be curtailed in yeast and other organisms [56,60–62]. We therefore explored the relationship between introgressions and the differences in CO and NCO counts between crosses. We modeled CO and NCO locations, correcting for GC content (a well characterized driver of recombination events [89,90]) and found that the distribution of COs and NCOs we observed could be partly explained by introgressions. While we are limited in our interpretations by only comparing two crosses (one cross with heterozygous introgression and one without introgression), these results are in line with findings in inversions, where heterozygotes show sharp decreases in COs, but the presence of NCOs in the inverted region [27,91]. However, unlike heterozygous inversions where an increase in COs is observed on freely recombining chromosomes (the inter-chromosomal effect), we do not see an increase in COs on the borders flanking intro-gression or on chromosomes without introgression. Relatedly, our results also differ from observations in *Arabidopsis*, in which COs are elevated in regions with more heterozygosity at the cost of COs in adjacent homologous regions [92,93]. These contrasting outcomes are likely in part due to the difference between intraspecific levels of polymorphism in *Arabidopsis* studies compared to the high level of inter-specific sequence divergence in our heterozy-gous introgression, although it also appears that mismatch repair gene *MSH2* has functionally diverged between *Arabidopsis* and *Saccharomyces* [94–96]. Regardless, it is curious that we do not find evidence for compensation of the depletion of COs due to introgression; the underly-ing mechanism responsible requires further study.

The introgressions present in the fermentation cross strains provided a unique opportunity to analyze NCO patterns, as the highly increased marker density lends the ability to detect NCOs with smaller tract lengths. We find evidence that DSB repair is biased towards NCOs when sequence similarity is low, with a weak but significant effect of increasing NCOs with decreasing sequencing similarity to a peak at sequence similarity between 0.8–0.9. Our data are thus aligned with previous work in *S. cerevisiae* and other systems in which the recombination intermediates D-loops or double Holliday junctions with mismatched sequences are disas-sembled, resulting in NCOs instead of COs [56,57,60–62,97,98]. By downsampling markers in the introgressed regions, we found an increase in the number of NCOs and a decrease in their tract length as we increased our resolution (S2–S10 Figs). Our expectation was to find a marker density at which NCO counts and their tract lengths would no longer increase and decrease respectively. However, we didn't find such a density, despite in some cases exploring marker densities that included all the markers in the introgression. This leaves us with the uncertainty of whether NCOs in the introgressions are more frequent and smaller than in other regions, or if we may be underestimating NCO counts and overestimating their tract lengths in cases where marker resolution is lower. Thus while a decrease in COs and increase in NCOs in heterozygous introgression compared to non-introgressed regions is consistent with prior work delineating the effect of sequence heterozygosity on DSB repair, our more limited resolution to detect NCO outside of introgression makes the NCO pattern more challenging to interpret.

One likely effect of the CO reduction we observed in introgressions is a reduction in allele shuffling at the regional and chromosomal level. While NCOs can increase local shuffling, they likely have a much weaker effect on the likelihood of two random alleles being shuffled than COs do. We find this is the case for our two crosses, where despite a large number of NCOs in introgressions, the amount of shuffling (as measured by $\bar{r}$ ) is significantly lower in

the fermentation cross. This loss of shuffling translates to frequently lower $\bar{r}$ in the fermentation cross at the chromosome level for chromosomes containing introgressions. The exceptions being small introgressions near the telomeres, which is consistent with the expectation that COs near the center of chromosomes generate much more shuffling of alleles than terminal COs [80]. Our findings indicate that reducing COs, especially near the center of chromosomes, has a cost to shuffling that is not compensated by the increase of NCOs that we observe. If the benefit of recombination is its ability to generate new combinations of alleles, then the loss of shuffling resulting from being heterozygous for divergent DNA sequences may come at an additional cost beyond the possibilities of genetic incompatibilities between hybridizing species. This cost is likely higher as divergence increases and as the length of divergent sequences is greater, as is the case with early generation hybrids [50]. Ultimately, if sequence divergence is too high, the resultant failure to recombine can become a postzygotic reproductive barrier [56,99,100].

The shuffling cost to introgression that we identify in our crosses may play an important role in the fate of introgression in the generations following hybridization. When heterozygotes for an introgression are formed, the reduction in shuffling inside the introgression will increase the likelihood that the introgression is purged from the population. This is because it will likely be inherited in its entirety and will carry the fitness cost of incompatibilities combined with a cost of shuffling. This cost is incurred because the reduction of COs in the introgression will reduce shuffling of alleles on either side of it and will vary in its intensity depending on the location and size of the introgression. In generations immediately following hybridization, introgressions will be much larger and are therefore expected to be more costly (although this likely depends on a number of factors including time since divergence). These predictions are consistent with modeling and empirical data on the purging of introgression in *Drosophila* and humans in the first generations following hybridization [44].

As to longer term dynamics of recombination and selection, we predict that the excess NCOs detected in heterozygous introgressions should begin to erode the divergence between the sequences, increasing homology and slowly reducing the cost of the introgression. This hypothesis posits that recombination can act to remove the larger, more deleterious regions of an introgression quickly while whittling away slightly deleterious alleles that may be linked to any beneficial regions of an introgression. While our current study doesn't capture longer term patterns of recombination or the landscape of recombination in introgressions that would lead to introgressions preferentially remaining in high-CO regions, it's notable that introgressed regions coincide with high-recombination regions in the natural cross, indicating that these regions may be primed to retain introgressions after a hybridization event–possibly due to the increased shuffling allowing for a larger fraction of the introgression to remain [44].

Finally, we note that *Saccharomyces* typically reproduce asexually, with sexual cycles occurring once every hundreds to thousands of mitotic generations [101–105]. When *Saccharomyces* strains do mate, they often mate within a tetrad resulting in increased homozygosity. For example, each diploid progenitor of the parents of our fermentation cross was homozygous for introgression across the genome, meaning that recombination would neither break up nor aid in purging the introgression in isolated populations of each parent. This suggests that the fate of introgressions in this species is perhaps more loosely tied to recombination patterns than it would be in an obligately sexually reproducing species.

Despite some limitations to interpretation, this study provides a unique view of the early dynamics of hybridization and the role of recombination in the presence of introgression. By focusing not only on the distribution of recombination events but on their specific role in shuffling alleles, we can more closely connect the physical process of recombination to its role among other evolutionary forces.

## Methods

### Strain and library construction

*S. uvarum* strains (UCD61-137, yHCT78, GM14, and DBVPG7787) were obtained from the Portuguese Yeast Culture Collection and from Chris Hittinger (S8 Table) [72]. All four *S. uvarum* strains had their *HO* locus replaced with a kanMX marker using a modified version of the high-efficiency yeast transformation using the LiAc/SS carrier DNA/PEG method. Briefly, the kanMX marker was amplified from plasmid pCSH2 with homology to genomic DNA flanking the *HO* ORF with primers CSH239 (GGTGGAAAACCACGAAAAGT TAGAACTACGTTCAGGCAAAgacatggaggcccagaatac) and CSH241 (GTGACCGTATTG GTACTTTTTTTGTTACCTGTTTTAGTAGcagtatagcgaccagcattc). For each strain, over-night cultures were inoculated in 25 mL of YPD at an OD of ~ 0.0005 and incubated at room temperature on a shaker for ~24 hours until the cultures reached an OD between 0.6 and 0.9. Subsequently, 1 ug of the template DNA was transformed with a heat shock temperature of 37°C for 45 minutes. The transformed cells were allowed to recover in liquid YPD for 4 hours before being plated onto G418 selective plates and incubated at room temperature for 2 days.

Single colonies were selected from the transformation plates, restreaked onto G418 plates and allowed to grow at room temperature for 2 days. Single colonies from those plates were then inoculated into 2 mL of YPD + G418 and incubated in a roller drum at room temperature overnight. From those cultures, 250 uL was used to inoculate 2 mL of sporulation media (1% potassium acetate, 0.1% yeast extract, 0.05% dextrose) and incubated at room temperature for 3 to 5 days. Strains were confirmed to have the ho::KanMX via tetrad dissection on a Singer SporPlay+ microscope (Singer Instruments). Plates with tetrads were incubated at room temperature for 2 days and then replica plated to test for proper segregation of the kanMX marker and mating type within individual tetrads.

Crosses between strains UCD61-137 and yHCT78 (natural cross), and between strains GM14 and DBVPG7787 (fermentation cross) were set up by micromanipulation of single *MATa* and *MATx* cells using a Singer SporPlay+. The plates were incubated at room temperature for 2 days and then replica plated to mating type tester strains to test for potential diploids. Identified diploids were then sporulated by growing a culture of the cross in 2 mL YPD + G418 at room temperature overnight. From those cultures, 250 uL were used to inoculate 2 mL of sporulation media and incubated at room temperature for 3 to 5 days. Sporulated cultures were dissected on 3 YPD plates (24 tetrads per plate) using a Singer SporPlay+. The spore viability of the natural cross was 93.4% and the spore viability of the fermentation cross was 89.93%. Fifty of the fully viable tetrads of each cross were selected and had all their spores inoculated into YPD (200 spores total per cross) and incubated at room temperature. The DNA was extracted from these cultures using a modified version of the Hoffman-Winston DNA Prep [106]. The DNA concentration was then measured using SYBR green, and 150 ng of each sample's DNA was used to prepare a sequencing library using an Illumina DNA Prep Kit, modified to use half the normal amounts of reagents. Libraries were pooled and run on an Illumina NovaSeq 500 with 150 bp paired end reads. The average sequence depth for the haploid spores is 41x coverage.

### Calling SNPs

We scored SNPs from parents and offspring using the *S. uvarum* reference genome [107] and custom scripts that invoked bwa (v0.7.17), samtools (v1.12), bcftools (v1.13), picard tools (v2.25.6), and gatk (v4.2.0.0) [108–110]. The custom scripts are available in the github repository: https://github.com/ejschwarzkopf/CO-NCO. We joint genotyped parents and offspring with default filters for gatk with the exception of the QUAL filter, which was set as < 100 for

parents and < 30 for offspring. We further filtered variants by requiring they be fixed differences between the two parental strains. We kept a total of 24,574 markers for the natural cross and 74,619 markers for the fermentation cross. We utilized LUMPY to identify structural variants in the parent strains that were greater than 5000 bp and verified calls using the Integrative Genomics Viewer [111,112]. We identified three amplifications in strain GM14 (one of the fermentation cross parents) that were absent in other strains (S9 Table).

## Generating CO/NCO maps

We generated "seg" files by coding tetrad variants by their parental origin. These seg files were the input for CrossOver (v6.3) from the ReCombine suite of programs, which we used to detect COs and NCOs [113]. We then filtered to remove non-crossovers with fewer than three associated markers and split the genome into 20kb windows. In each window we counted crossovers, non-crossovers, and markers. We established regions of introgressions through visual inspection of marker density in the fermentation cross (introgressions showed more divergence between fermentation strains) and confirmed them using the findings of Almeida et al. [72]. We found nine heterozygous introgressions on chromosomes 4, 6, 7, 9, 10, 10, 13, 14, and 15 respectively that we included in further analyses (S10 Table). We excluded two additional introgressions due to poor mapping (chromosome 13:0–17,000; chromosome 16: 642,000–648,000). We analyzed the effect of marker resolution on NCO detection by randomly removing markers from each introgression and running CrossOver and its downstream counting process as described above. For each introgression, we downsampled to multiple percentiles of the distribution of non-introgressed region marker counts per kb (10th: 0.05; 25th: 0.75; 50th: 1.95; 75th: 2.95; 90th: 4.45; 95th: 7.87; 99th: 14.01) and the median marker count per kb for introgressed (3.225) and non-introgressed (1.35) regions. We repeated each downsampling amount 20 times for each introgression and extracted average CO counts, NCO counts, and NCO tract lengths. To account for the difference in number of markers in introgressed vs non-introgressed windows and their effect on NCO detection, we applied a previously published simulation-based method [33,67]. We chose the average NCO tract length of the fermentation cross's introgressed regions–550 bp–and for each window randomly placed an NCO event of that length 10,000 times to establish our expected probability of detecting an NCO of that length. We then divided our observed NCO count by our probability of detecting an NCO event. Additionally, because COs that occurred in large regions devoid of markers would be called in the middle of the empty windows, we split CO counts in regions with multiple consecutive marker-free windows evenly between the empty windows. With these corrected maps, we calculated spearman correlations between crosses using R (v4.1.0) [114]. Additionally, we modeled NCO and CO count as a function of introgression, introgression by cross, and GC content using a gaussian generalized linear model in R (v4.1.0) [114].

## Sequence similarity

We calculated diploid sequence similarity between the two fermentation cross strains in 101 bp windows with 50 bp overlaps. At each nucleotide position in the window, we counted fixed differences as zero diploid sequence similarity, invariant sites between strains as full diploid sequence similarity (1), and polymorphic sites in either or both strains as half diploid sequence similarity (0.5). We then averaged these sequence similarity values across the window. This measure represents the probability that both strains will have the same nucleotide base at a given position. We used this measure of fine-scale sequence similarity to determine how sequence similarity related to NCO counts in introgressed regions. For this, we used

Loess regressions and Spearman's correlations on each of the introgressed regions comparing sequence similarity to NCO count, both implemented in R (v4.1.0) [114]. We then focused on each recombination event (CO or NCO) and compared the sequence similarity 100 bp up and downstream of CO breakpoints and 100 bp up and downstream of NCO tracts. We then used Welch's two sample t-tests to compare CO and NCO sequence similarity in each introgression.

## $\bar{r}$

We use $\bar{r}$, a measure genetic shuffling defined in Veller et al. [80] to measure how much shuffling occurs in each chromosome for each cross. Our data provides parental origin for each fixed difference between parental strains. We assume that all loci between pairs of markers that come from the same parent are also from that parent. We also assume that when a pair of successive markers come from different parents, the location of the change from one parental origin to the other happens at the midpoint between our markers. With this in mind, we counted the number of bases that come from one parent and divided by the chromosome size to obtain the proportion of the chromosome that was inherited from said parent ($p$) and used the formula from Veller et al. [80]: $\bar{r} = 2p(1-p)$. We calculated $\bar{r}$ for each full chromosome and each introgressed regions in every gamete from both crosses. We then averaged across gametes to obtain average $\bar{r}$ values. We then compared average $\bar{r}$ between crosses in each chromosome or introgressed region using Welch two sample t-tests and correcting for multiple tests using a Bonferroni correction in R (v4.1.0) [114].

## Supporting information

**S1 Fig. *S. uvarum* chromosomes not containing introgressions split into 20kb, non-overlapping windows.** CO, NCO, and SNP counts are reported for both crosses (fermentation and natural). CO counts are smoothed when the true location of the CO split could be in one of multiple windows. NCO counts are corrected for marker resolution.
(EPS)

**S2 Fig. Downsample densities and NCO rate, CO rate, and tract length. Introgression on chromosome 4.**
(EPS)

**S3 Fig. Downsample densities and NCO rate, CO rate, and tract length. Introgression on chromosome 6.**
(EPS)

**S4 Fig. Downsample densities and NCO rate, CO rate, and tract length. Introgression on chromosome 7.**
(EPS)

**S5 Fig. Downsample densities and NCO rate, CO rate, and tract length. Introgression on chromosome 9.**
(EPS)

**S6 Fig. Downsample densities and NCO rate, CO rate, and tract length. First introgression on chromosome 10 (10a).**
(EPS)

**S7 Fig. Downsample densities and NCO rate, CO rate, and tract length. First introgression on chromosome 10 (10b).**
(EPS)

**S8 Fig. Downsample densities and NCO rate, CO rate, and tract length. Introgression on chromosome 13.**
(EPS)

**S9 Fig. Downsample densities and NCO rate, CO rate, and tract length. Introgression on chromosome 13.**
(EPS)

**S10 Fig. Downsample densities and NCO rate, CO rate, and tract length. Introgression on chromosome 15.**
(EPS)

**S11 Fig. Loess regression plots of NCOs as a function of homology in the introgressions of the fermentation cross.** The left column's y-axis is scaled to the size of the loess curve, while the right column's y-axis is scaled to the NCO count. The gray shading indicates the standard error for the loess estimates. Chromosome 14's SE ribbon was removed for readability.
(EPS)

**S1 Table. Mean and standard error of CO counts per chromosome for fermentation and natural crosses.**
(DOCX)

**S2 Table. Mean and standard error of NCO counts per chromosome for fermentation and natural crosses.**
(DOCX)

**S3 Table. Summaries of linear regression for downsampled introgressions.** Each regression has either CO count, NCO count or NCO tract length as the response variable, and marker density as the independent variable.
(DOCX)

**S4 Table. Spearman's correlations between crosses in non-introgressed regions.**
(DOCX)

**S5 Table. Spearman's correlations between crosses in introgressed regions.**
(DOCX)

**S6 Table. $\bar{r}$ values for whole chromosomes.**
(DOCX)

**S7 Table. $\bar{r}$ values for introgressed regions.**
(DOCX)

**S8 Table. Strain information.**
(DOCX)

**S9 Table. Copy number variation in GM14.**
(DOCX)

**S10 Table. Location of introgressions.**
(DOCX)

## Acknowledgements

We are grateful to members of the Heil lab, Mohamed Noor, Nathan Layman, and Mark Smithson for comments on this manuscript. We thank Chris Hittinger and the Portuguese Yeast Culture Collection for *S. uvarum* strains.

## Author contributions

**Conceptualization:** Caiti Smukowski Heil.

**Data curation:** Enrique J. Schwarzkopf.

**Formal analysis:** Enrique J. Schwarzkopf.

**Funding acquisition:** Caiti Smukowski Heil.

**Investigation:** Enrique J. Schwarzkopf, Nathan Brandt.

**Methodology:** Enrique J. Schwarzkopf.

**Project administration:** Caiti Smukowski Heil.

**Resources:** Nathan Brandt, Caiti Smukowski Heil.

**Supervision:** Caiti Smukowski Heil.

**Validation:** Enrique J. Schwarzkopf.

**Visualization:** Enrique J. Schwarzkopf.

**Writing – original draft:** Enrique J. Schwarzkopf, Caiti Smukowski Heil.

**Writing – review & editing:** Enrique J. Schwarzkopf, Caiti Smukowski Heil.

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
