## [Decision Letter · Decision Letter 0]

28 Oct 2024

PGENETICS-D-24-01107The recombination landscape of introgression in yeastPLOS Genetics Dear Dr. Smukowski Heil, Thank you for submitting your manuscript to PLOS Genetics. After careful consideration, we feel that it has merit but does not fully meet PLOS Genetics's publication criteria as it currently stands. Therefore, we invite you to submit a revised version of the manuscript that addresses the points raised during the review process. Please submit your revised manuscript within 60 days Dec 27 2024 11:59PM. If you will need more time than this to complete your revisions, please reply to this message or contact the journal office at plosgenetics@plos.org. Please include the following items when submitting your revised manuscript:* A rebuttal letter that responds to each point raised by the editor and reviewer(s). You should upload this letter as a separate file labeled 'Response to Reviewers '. This file does not need to include responses to any formatting updates and technical items listed in the 'Journal Requirements' section below.* A marked-up copy of your manuscript that highlights changes made to the original version. You should upload this as a separate file labeled 'Revised Manuscript with Track Changes '.* An unmarked version of your revised paper without tracked changes. You should upload this as a separate file labeled 'Manuscript '. If you would like to make changes to your financial disclosure, competing interests statement, or data availability statement, please make these updates within the submission form at the time of resubmission. Guidelines for resubmitting your figure files are available below the reviewer comments at the end of this letter. We look forward to receiving your revised manuscript. Kind regards, Jun-Yi LeuAcademic EditorPLOS Genetics Justin FaySection EditorPLOS Genetics

Aimée Dudley

Editor-in-Chief

PLOS Genetics

Anne Goriely

Editor-in-Chief

PLOS Genetics

**Journal Requirements:** **Additional Editor Comments (if provided):** As you can see, Reviewer 2 raised several concerns about the experiments, especially the differences in SNP numbers between crosses and the efficiency of NCO calling due to the differences. I agree that computer simulation with marker density downsampling can partially address this issue, but it still differs from actual experimental data. I feel it is necessary to discuss this caveat in detail so readers can be aware of it when reading your data. Including one other non-introgression cross will make your conclusions more solid, but I will leave it to you to decide.**Reviewers' comments:** Reviewer's Responses to Questions

**Comments to the Authors:**

Reviewer #1: This is a simple but interesting and well executed study using high-throughput sequencing to look at the effect of DNA sequence differences in homologous chromosomes on recombination. It’s been known for decades that crossing over is less likely to occur in regions where there are differences in sequence than where sequences are identical, but how this plays out in the overall number of crossover and non-crossover recombination events across whole genomes is still not entirely clear.

Here the authors use two S. uvarum yeast crosses, one formed between two natural strains, with about 24,000 SNP differences overall, and one formed between two brewery strains, one of which contained a few introgressions from another species (S. eubayanus) introducing a higher level of divergence at the introgressed regions; this cross had 74, 000 SNPs overall. This design meant that comparisons could be made not only between the crosses, but also between the introgressed and non-introgressed regions of the brewery cross, with the equivalent regions in the wild cross (which lacked introgressions) serving as controls. The authors sequenced 48 complete tetrads from the two crosses, and mapping all their crossover (CO) and non-crossover (NCO) events.

The results are well presented and analysed, so well that there’s no need to repeat them except very briefly. Overall they are in line with expectations, that more heterozygous regions are more likely to get NCOs and less likely to get COs. Because the whole genome was sequenced we see not only how the crossover density changes in certain regions, but also how the total number is affected (or not). The authors carefully consider the different factors affecting their results, and perform effective analysis and controls. The discussion is interesting and thorough and the writing is excellent.

Unusually, I can’t find much to criticise here. I couldn’t think of anything the authors had missed. I liked that the figures are effective and the most useful are all in the main text, rather than in the supp. The best I can come up with is a lack of italisation in places, e.g. please search for MSH2, Saccharomyces, Arabidopsis, and Drosophila. I’m sure the proof reader would have found these anyway. So I recommend acceptance without major revision. Congratulations to the authors on a nice piece of work.

Reviewer #2: In this study the authors sporulated two crosses of S. uvarium. One cross contained an introgression with S. eubayanus (labeled as fermentation) and the other lacked the introgression (natural). The authors analyzed 9 introgressed regions in the fermentation strain and noted that they were ~6% divergent from the syntenic region in S. uvarium, covering ~10% of the genome. Four-viable tetrads from 48 meioses (in total or for each strain?) were analyzed by Illumina 150 bp paired end DNA sequencing. Crossover (CO) and noncrossover (NCO) events were scored based on 24,574 SNP differences in the natural cross and 74,619 in the fermentation cross. The authors used a filter to remove potential NCO events in which there were fewer than three associated markers. They also performed simulations and statistical analyses to account for the different SNP populations in the two strains. The main conclusions were:

i. The fermentation cross had a lower average CO count per meiotic cell (64) compared to the natural cross (83). The introgressed regions in the fermentation cross showed particularly low CO counts compared to their corresponding syntenic regions in the natural cross.

ii. The frequency of NCOs appeared higher in the introgressed regions. An increase in NCOs in the fermentation cross was seen for introgressed regions as similarity was reduced (until about 0.9 to 0.8 sequence similarity) but the overall correlation was modest. The authors performed a set of statistical measures to determine the impact of increased NCOs on gene shuffling; they concluded that the NCO effect on local shuffling did not appear to help shuffling in the fermentation cross.

iii. In the non-introgressed regions of the genome CO levels in the natural cross were higher than in the fermentation cross (Fig 2A). One possibility is that non-introgressed regions in the natural cross are predisposed to having more recombination though there are certainly other explanations.

Comments

To my knowledge this is the first detailed analysis of meiotic CO and NCO events using an introgressed cross in Saccharomyces. There are similarities between the submitted work and work by Cooper et al. (https://doi.org/10.1101/480418) who observed in hybrid S. cerevisiae crosses that COs tended to occur away from polymorphic regions in DNA mismatch repair proficient cells. Based on analyzing strains defective in Class I CO promoting factors and the Msh2 mismatch repair protein as well as modeling analyses of their mutant phenotypes, Cooper et al. argued that Class I COs are more sensitive to heteroduplex DNA that forms during recombination. In support of this idea, Dash et al. (Genetics 226, iyad214) found that Msh5 protein, which is part of the Msh4-Msh5 complex which acts in the Class I crossover pathway, showed reduced localization in regions of higher sequence divergence, suggesting that Msh4-Msh5 displays lower levels of binding to recombination intermediates that contain higher levels of mismatches. Together these studies suggest that even low levels of sequence divergence can cause significant changes in CO frequency, distribution and type. The authors are aware of these observations and cite them.

A challenge for this reviewer is that the natural and fermentation crosses involve different numbers and presumably genomic locations of SNPs. I recognize that it would not be impossible to have isogenic crosses which only differ by introgression, but as noted by the authors CO numbers can vary significantly within a single organism. Also, CO levels in the natural strain were higher than in the fermentation strain in the non-introgressed regions of the genome (representing ~90% of the genome, Fig 2A). The work done in S. cerevisiae with hybrid strains shows that relatively low sequence divergence can alter CO patterns; perhaps I did not look carefully enough but I could not find information where the authors compared the SNP densities in the two crosses and asked (outside of the introgressed regions) whether they could impact CO distribution. I recognize that the authors are very cautious in their interpretations and work hard to buttress their arguments through statistical analyses, but the key takeaway from this work for this reviewer is the effect of introgression on CO distribution, with the other observations being less clear. The CO effect is not surprising based on the above work done in S. cerevisiae, but COs in different organisms (as stated by the authors) appear to respond differently to sequence divergence (see studies in mouse-Peterson et al., Mol Cell 78:1252, and Arabidopsis-Blackwell et al., EMBO J 39: e104858). This is perhaps an unreasonable ask, but including one other non-introgression cross would provide evidence that some of the conclusions made in this study cannot be explained by just cross-specific variation in CO levels.

The observations about NCO events in the natural and fermentation crosses are hard to interpret. I agree with Cooper et al. that it may be difficult to make strong conclusions about NCO levels because NCOs are impacted to a greater degree by crossover homeostasis (see Martini et al. Cell 126:285) than COs. It’s also relatively easy to identify a CO whereas as Cooper et al. note: “NCOs are directly affected by “the true number of converted and/or heteroduplex markers contained within a NCO event and by the technical efficiency of calling what could be only short regions of contiguous nonreciprocal marker exchange.” The fact that the authors have different SNP numbers/distributions in the two crosses is likely to make NCO calls even more challenging.

Minor issues.

It would be valuable for the authors to include in the main text the average read depth for the Illumina sequencing, and the spore viability of the natural and fermentation crosses.

Line 87. The authors state: “Much of the evolutionary advantage of recombination is understood to originate from its role in shuffling alleles…”. It might be worth indicating here that independent assortment of chromosomes appears to have a much greater effect on genetic shuffling than crossing over. One recent study suggested that independent assortment contribution is roughly 30 times greater than that provided by crossing over (Veller et al. Proc. Natl. Acad. Sci. USA 116:1659).

Lines 95 to 101, 146, 290. In the last 20 years a significant amount of work has led to the identification of two different CO pathways in S. cerevisiae meiosis. The Class I pathway, which represents ~85% of meiotic crossovers in S. cerevisiae, result from the biased resolution of double-Holliday junctions to exclusively form crossovers that display interference. The remaining COs occur through the Class II pathway; this pathway is thought to result from the unbiased resolution of double-Holliday junctions to form COs and NCOs. Including a brief overview of this literature would be valuable, especially since studies cited above have suggested that the Class I CO pathway in S. cerevisiae is impacted by sequence divergence.

Line 106. “NCOs are also crucial to reducing linkage within coding regions and unlike COs, result in 3:1 allele ratio in the meiotic product…. It’s important to note that in most DSBR models both COs and NCOs are associated with gene conversion events.

Line 139. “A DSB in a region of low sequence similarity will recruit mismatch repair proteins…” This text suggests that the DSB already knows that it is in a region of low sequence similarity and will recruit mismatch repair proteins before it is processed into a heteroduplex intermediate.

Line 451. The authors state that “…Saccharomyces typically reproduce asexually with only infrequent sexual cycles…” Yes this is what has been observed in the cited studies, but there are arguments in the literature (reviewed in Marsit and Dequin, FEMS Yeast Res. 15:fov067) suggesting that outcrossing could increase significantly in stressful environments, perhaps to one in every 100 to one in every two generations.

**Have all data underlying the figures and results presented in the manuscript been provided?**

Reviewer #1: Yes

Reviewer #2: Yes

PLOS authors have the option to publish the peer review history of their article (what does this mean? ). If published, this will include your full peer review and any attached files.

**Do you want your identity to be public for this peer review?** For information about this choice, including consent withdrawal, please see our Privacy Policy .

Reviewer #1: No

Reviewer #2: No

---

## [Editor Report · Decision Letter 1]

21 Jan 2025

Dear Dr Smukowski Heil,

We are pleased to inform you that your manuscript entitled "The recombination landscape of introgression in yeast" has been editorially accepted for publication in PLOS Genetics. Congratulations!

Yours sincerely,

Jun-Yi Leu

Academic Editor

PLOS Genetics

Justin Fay

Section Editor

PLOS Genetics

Aimée Dudley

Editor-in-Chief

PLOS Genetics

Anne Goriely

Editor-in-Chief

PLOS Genetics

Comments from the reviewers (if applicable):

**Data Deposition**

http://datadryad.org/submit?journalID=pgenetics&manu=PGENETICS-D-24-01107R1

**Press Queries**

---

## [Editor Report · Acceptance letter]

PGENETICS-D-24-01107R1

The recombination landscape of introgression in yeast

Dear Dr Smukowski Heil,

We are pleased to inform you that your manuscript entitled "The recombination landscape of introgression in yeast" has been formally accepted for publication in PLOS Genetics! Your manuscript is now with our production department and you will be notified of the publication date in due course.

With kind regards,

Anita Estes

PLOS Genetics

On behalf of:
